# Estimating population size when individuals are asynchronous: A model illustrated with northern elephant seal breeding colonies

**Richard Condit**[1]*, **Sarah G. Allen**[2], **Daniel P. Costa**[1], **Sarah Codde**[2], **P. Dawn Goley**[3], **Burney J. Le Boeuf**[1], **Mark S. Lowry**[4], **Patricia Morris**[1]

**1** Institute for Marine Sciences, University of California, Santa Cruz, Santa Cruz, California, United States of America, **2** National Park Service, Point Reyes National Seashore, Point Reyes Station, CA, United States of America, **3** Department of Biological Sciences, Humboldt State University, Arcata, CA, United States of America, **4** Marine Mammal and Turtle Division, Southwest Fisheries Science Center, La Jolla, CA, United States of America

* conditr@gmail.com

**Data Availability Statement:** Complete census data, including all daily female counts, are

## Abstract

Our aim was to develop a method for estimating the number of animals using a single site in an asynchronous species, meaning that not all animals are present at once so that no one count captures the entire population. This is a common problem in seasonal breeders, and in northern elephant seals, we have a model for quantifying asynchrony at the Año Nuevo colony. Here we test the model at several additional colonies having many years of observations and demonstrate how it can account for animals not present on any one day. This leads to correction factors that yield total population from any single count throughout a season. At seven colonies in California for which we had many years of counts of northern elephant seals, we found that female arrival date varied < 2 days between years within sites and by < 5 days between sites. As a result, the correction factor for any one day was consistent, and at each colony, multiplying a female count between 26 and 30 Jan by 1.15 yielded an estimate of total population size that minimized error. This provides a method for estimating the female population size at colonies not yet studied. Our method can produce population estimates with minimal expenditure of time and resources and will be applicable to many seasonal species with asynchronous breeding phenology, particularly colonial birds and other pinnipeds. In elephant seals, it will facilitate monitoring the population over its entire range.

## Introduction

Although knowledge of the breeding population of any species is fundamental for theoretical ecology and conservation, assembling counts of all individuals is seldom possible. The problem that arises, therefore, is estimating what proportion of a population can be counted [1]. One aspect of this problem is asynchrony, leading to a situation in which there is no one time when the entire population is present in a study area [2, 3]. This problem can be solved with modern

published as a Dryad Data Archive (https://doi.org/10.7291/D1PP47). The software is available online, where it can be applied to any input data (http://conditdatacenter.org/sealcensus/estimator/), and source code is posted at Github (https://github.com/richardcondit/sealcensus).

**Funding:** The research was funded by numerous grants over 50 years, some to every one of the authors, from National Science Foundation, Office of Naval Research, Bureau of Land Management, National Marine Fisheries Service, National Geographic Society, National Park Service, San Francisco Bay Area Network Inventory and Monitoring Program, Point Blue Conservation Science, Point Reyes Bird Observatory, Point Reyes National Seashore provided financial support, and the University of California, Santa Cruz. The funders had no role in study design, data collection and analysis, decision to publish, or preparation of the manuscript.

**Competing interests:** The authors have declared that no competing interests exist.

mark-recapture tools, often revealing that direct counts are substantial underestimates of the total population [3]. These superpopulation methods, however, require marked individuals that can be reidentified on several surveys. Since this is a substantial hindrance in many circumstances, we present here an alternative method for estimating total population in asynchronous species from single counts. Our approach is based on some prior knowledge of individual behavior, but once that is collected, the model works with counts but requires no marked individuals. We first developed the model [4] using observations of colonies of the northern elephant seal (*Mirounga angustirostris*).

Elephant seals are large marine predators that aggregate on beaches to reproduce at predictable times each year [5, 6]. Females come ashore to raise pups and to mate, and their dense groups can be approached and easily counted. Males are also present, but because females each produce a single pup, the most important index of population size is the number of breeding females. Moreover, because most of the population is found in just eight colonies [7], only a few sites need to be counted to generate estimates of the entire population. The problem of asynchrony arises, however, because there is never a day when all breeding individuals are ashore. While the entire breeding season lasts nearly three months, individual females are on shore only one month: some depart before others have arrived. Our goal is determining what fraction of the females are missed in any one count. To do so, we make use of a published model whose purpose was to quantify asynchrony in elephant seals [4].

The model uses observed daily counts to estimate the timing of arrival and departure of females at a colony [4, 8, 9]. This quantifies female asynchrony and leads to an estimate of the number of animals missed during any one count. The present objective is to apply the model to several of the largest northern elephant seal colonies where females have been counted repeatedly. We quantify consistency of the timing of female arrival, over many years and between sites, and use the model's estimate of the fraction of females onshore on any one day to generate correction factors that convert a single count to total female population. We also provide an online software tool that applies the model to any input data, allowing the method to be tested in other asynchronous species such as colonial birds or pinnipeds.

## Materials and methods

### Ethics statement

Seal observations were authorized under permits 939, 373–1575, 373–1868-00, 17152–00, 2142514535, 14535, 14636, and 21425 from the National Marine Fisheries Service, and Marine Mammal Protection Act Permit 486. Access to park land was granted by the California Department of Parks and Recreation.

### Field sites

We used daily censuses of breeding females at seven northern elephant seal colonies (Fig 1), from 1968–2018 at Año Nuevo Island, 1977–2018 at Año Nuevo Mainland, 1981–2019 at Point Reyes, 2018–2020 at King Range, plus 2010 and 2013 at San Nicolas Island and 2013 at San Miguel Island and Santa Rosa Island. These colonies include the three largest northern elephant seal colonies in the world and 70% of the total population.

### Female counts

In what follows, all data are counts of all females onshore at one colony on one day. At Año Nuevo, we usually counted from high points on dunes near the females [6], but counts from airplanes or drones were used to support ground counts on a few days when numbers were at

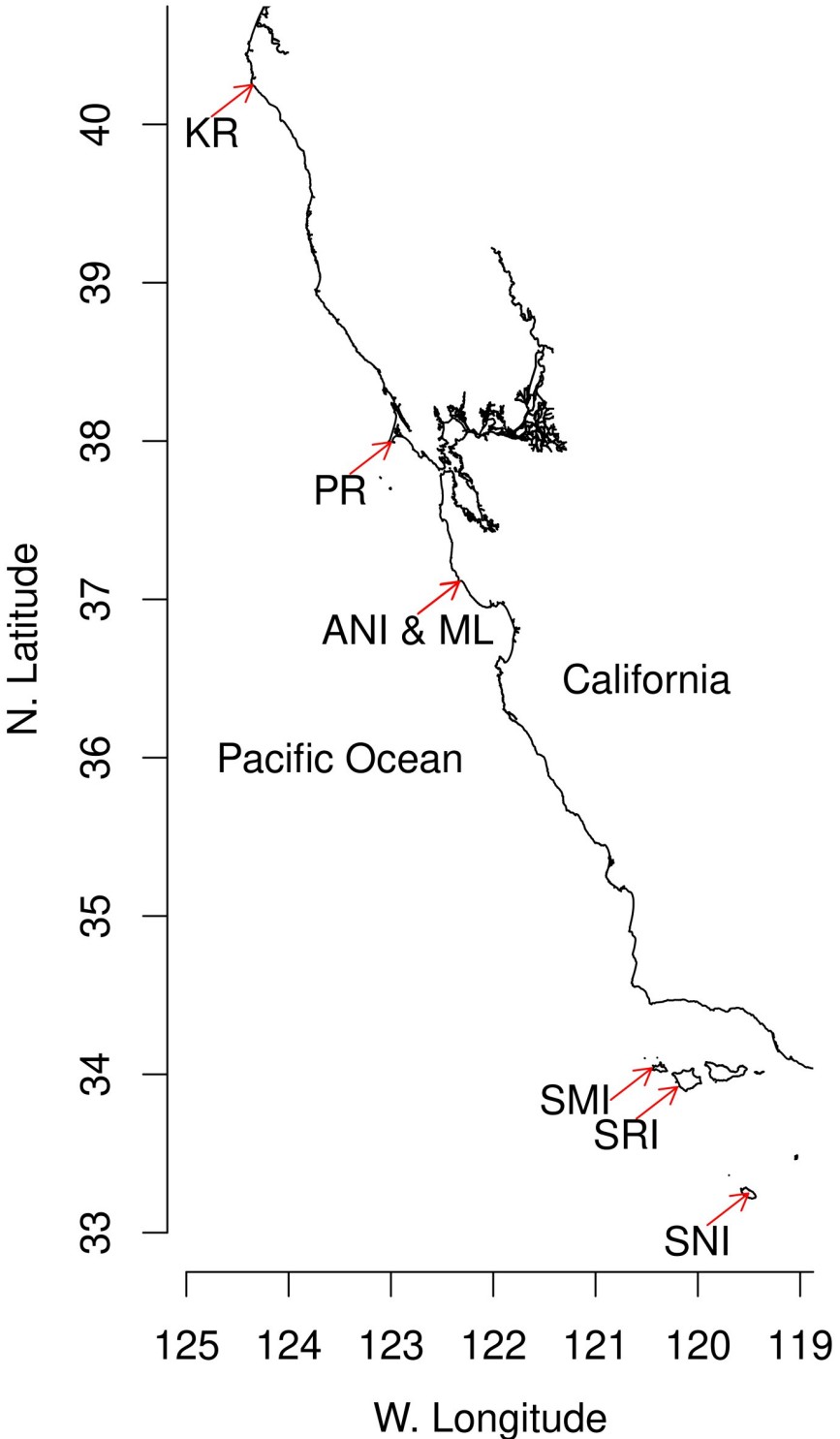

**Fig 1. Northern elephant seal breeding colonies.** Locations of the seven colonies analyzed, indicated by red arrows: San Nicolas Island (SNI), Santa Rosa Island (SRI), San Miguel Island (SMI), Año Nuevo island and mainland (ANI & ML), Point Reyes (PR), King Range (KR).

their highest. Aerial counts confirmed that ground counts had no consistent bias. Most counts at Point Reyes were done from high cliffs overlooking the animals, allowing excellent visibility, but some were done on beaches immediately adjacent to the females [10]. Counts at King Range were done from close proximity to the colony, always within 100 m of the animals [11]. At the Channel Islands, all counts were made from aerial photographs and have been verified elsewhere [12, 13].

## Census model

We define the census curve, $C(t)$, as the count through time during one season at one colony. In elephant seals, it is bell-shaped, as in other seasonal breeders (eg [14, 15]); at Año Nuevo, it starts from zero in mid-December, peaks near the end of January, then declines back to zero by early March (ref [4]; Fig 2). Define arrival $a(t)$ and departure $d(t)$ as the total number of

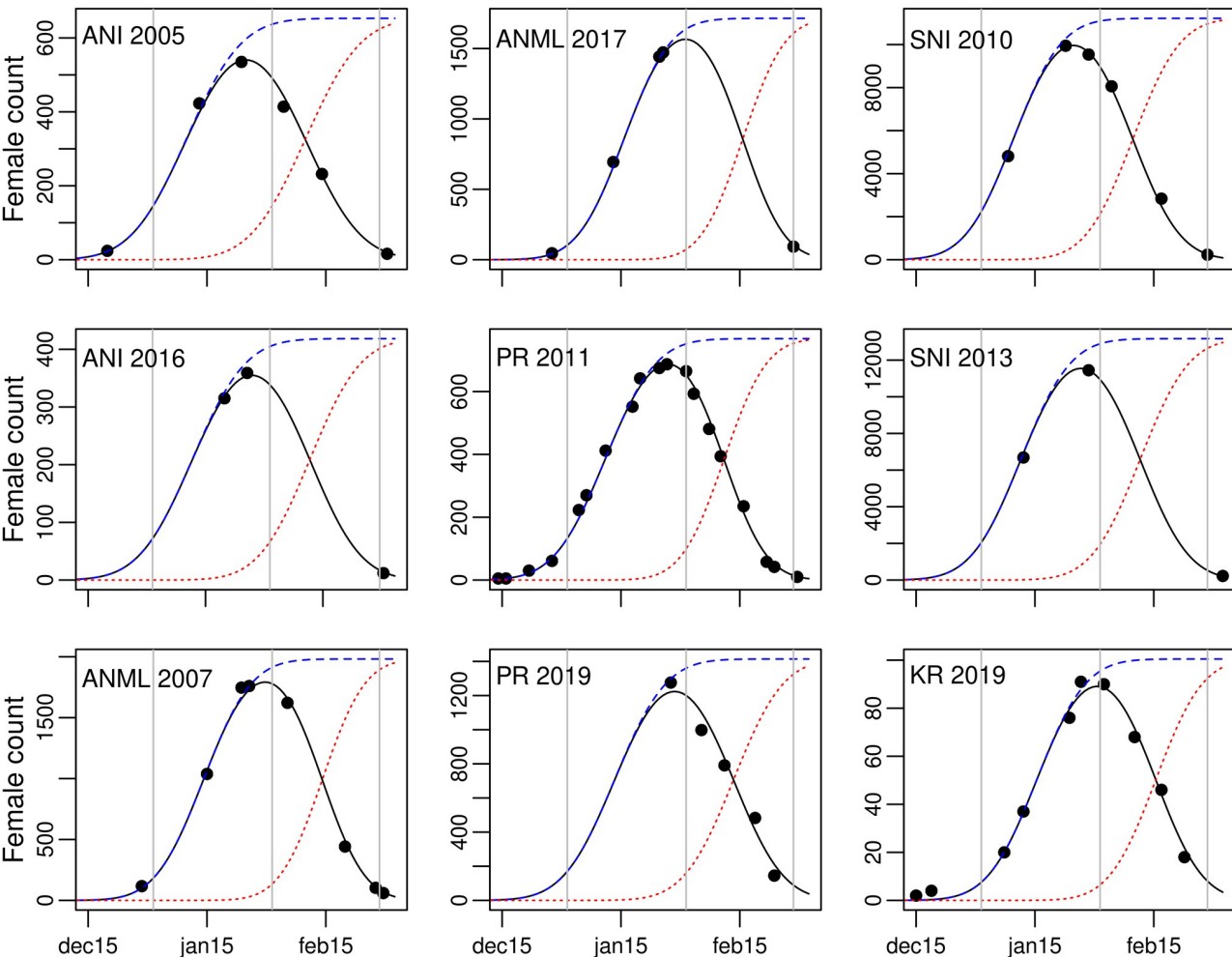

**Fig 2. Daily female counts and fitted census curves in sample years.** Black points are observed counts on single days. The black, bell-shaped curve is the fitted census curve, $C(t)$; the blue dotted curve is the estimated cumulative arrival curve, $A(t)$; and the dotted red curve the cumulative departure curve, $D(t)$. $A(t)$ and $D(t)$ reach an asymptote at the total population of females, $N_i$, using the colony in year $i$. The examples include years with few counts, demonstrating that the model works well with smaller samples, plus sample years from Point Reyes, King Range, and the Channel Islands; these complement our earlier presentation of the model [4] in which we included similar figures for 18 years at Año Nuevo Mainland, 1978–2006. Abbreviations: ANI, Año Nuevo Island; ANML, Año Nuevo Mainland; PR, Point Reyes; SNI, San Nicolas Island; KR, King Range. Range of the vertical axis varies greatly, but the dates on the horizontal axis are identical, with vertical lines at 1 Jan, 1 Feb, 1 Mar.

**Table 1. Six parameters for the census model in a single year and their hyper-parameters across years.** The six main parameters are subscripted since there is one for every year $i$; each colony has a separate set. The population size had no hyper-parameters, meaning every year's population was independent, and the two parameters describing tenure had hyper-parameters fixed in advance as narrow priors ($\mu_d = 31.06$, $\sigma_d = 0.27$, $\mu_v = 0.1212$, $\sigma_v = 0.0062$), a requirement for fitting the model [4]. The remaining parameters had Gaussian hyper-distributions estimated from the data. In a model covering $Y$ years of censuses, $6Y + 10$ parameters are needed: $6Y$ main parameters (6 per year) plus 10 hyper-parameters.

| Parameter | Annual | Hyper-parameter | |
|---|---|---|---|
| | | Mean | SD |
| Female population | $N_i$ | none | none |
| Mean arrival date | $\hat{a}_i$ | $\mu_a$ | $\sigma_a$ |
| SD arrival date | $s_i$ | $\mu_s$ | $\sigma_s$ |
| Correlation arrival-tenure | $c_i$ | $\mu_c$ | $\sigma_c$ |
| Mean tenure on colony (Prior) | $\hat{d}_i$ | $\mu_d$ | $\sigma_d$ |
| CV tenure (Prior) | $v_i$ | $\mu_v$ | $\sigma_v$ |

animals arriving (departing) on each day $t$, then cumulative arrival $A(t) = \sum_1^t a(i)$ and departure $D(t) = \sum_1^t d(i)$ represent the number of females that arrived (departed) through day $t$. The estimated census curve on day $t$ is

$$C(t) = A(t) - D(t). \tag{1}$$

Our model describes $a(t)$ and $d(t)$ as Gaussian functions of $t$, and the parameters describing those curves (Table 1) are estimated by fitting $C(t)$ as closely as possible to the observed daily counts (Fig 2). In northern elephant seals, the model generated census curves $C$ that closely fit observed counts [4], and the same approach has been applied in southern elephant seals [8, 9]. Both cumulative arrival and departure curves, $A(t)$ and $D(t)$, reach an asymptote at the total population, $N$, so once parameters for $a$ and $d$ are estimated, the fraction of animals missed in any one count $C$ can be calculated. Further details are presented in Condit et al. [4].

Here we add one substantial improvement. In the previous study [4], we estimated a set of parameters separately in each year. Here, we fit the entire ensemble of counts at one colony in a single hierarchical model, with year as a random effect [16]. Taking the mean arrival date, $\hat{a}$, as an example, each year $i$ has an estimated $\hat{a}_i$, and those dates across years are fitted to a Gaussian distribution known as the hyper-distribution, described by a hyper-mean arrival date, $\mu_a$, and a hyper-standard-deviation, $\sigma_a$ (Table 1). Because we hypothesize some year-to-year consistency in the timing of breeding, the 5 parameters describing timing were constrained by a Gaussian hyper-distribution (Table 1), but we have no expectation of consistency in population size, so $N_i$ was not constrained. The advantage of the hierarchical method arises in years having too few counts to fit the full arrival-departure model. In previous papers, we omitted those years from calculations [4, 6]. With a multi-year model, they can be included because the arrival and departure curves from other years constrain the arrival timing (Fig 2).

The multi-year hierarchical model was fitted to five sets of data: Año Nuevo Island, Año Nuevo Mainland, Point Reyes, King Range, and the three Channel Islands combined; data were insufficient to model the Channel Islands separately. We did not attempt to create a single grand model with both colony and year levels, since there were too few colonies to support a random effect. All parameters of the model (Table 1) for those five datasets were estimated with a Bayesian Monte Carlo procedure, producing a posterior distribution and thus 95% credible intervals for parameters and the statistics calculated from parameters; details on

parameter-fitting are given in [4, 17, 18]. To compare colonies and to evaluate consistency of the census timing across years, we used the mean arrival date each year, $\hat{a}_i$, as fitted by the model, and the date of the maximum female census calculated from the fitted census curve, $C(t)$, in each year. Non-overlapping 95% credible intervals were used to infer statistically significant differences.

## The correction factor

Consider site $s$ in year $i$ and define the number we seek as $N_{is}$, the total number of breeding females using the site in that year. $C_{is}(t)$ is the census curve at site $s$ in year $i$: the estimated daily female population on each day $t$ (Eq 1, Fig 2). Then $C_{is}(t)/N_{is}$ is the proportion of females on the colony on day $t$. The inverse is the correction factor, a multiplier for the female count on day $t$ that yields the total female population,

$$m_{is}(t) = \frac{N_{is}}{C_{is}(t)}. \tag{2}$$

The correction factor was calculated separately in every year at a site, but to be useful in future years, we need the average across all years for site $s$, $\hat{m}_{is}(t)$. To generate $\hat{m}_{is}(t)$, census curves for each site $s$ were simulated using random draws of parameters from the joint posterior distributions at that site; the draws included the 5 parameters describing timing, while population size was set at $N = 1000$. Each simulation produced one $m_s(t)$, a correction factor for each day $t$ at site $s$. To capture the total error associated with a multiplier, we incorporated both year-to-year variation and within-year error. For the former, we sampled 30 years at random from each site, and for the latter, made 40 random draws from the parameters' posterior distributions within each year; the mean and 95% quantiles of those 1200 simulations yield $\hat{m}_{is}(t)$. To be most relevant for the present, years were chosen from 2002–2018 at Año Nuevo Island, Año Nuevo Mainland, and Point Reyes. This was a period in which censuses and the model fit were consistent and reliable. There were only four years available at the Channel Islands and three at King Range, so all were utilized.

## Results

### Census timing

Our model converged on estimates of total population size and arrival behavior at all northern elephant seal colonies we studied in every season. Observed counts and fitted census curves followed a bell-shaped curve from late December to early March at all sites (Fig 2). The model quantified year-to-year variation in arrival dates and thus the day of the maximum number of animals present (Fig 3), and it generated consistent census curves in several years having few counts (Fig 2).

Comparison of arrival date and peak census date across years and across sites illustrates the degree of consistency (Fig 3). In early years at Año Nuevo Mainland, arrival dates were around 17 Jan, but were consistently near 14 Jan after 1990. Arrival dates at Point Reyes and Año Nuevo Island also started relatively late, then stabilized 3–4 days earlier (Fig 3A). The standard deviation in arrival date between years ($\sigma_a$) was 1.8–1.9 days at Año Nuevo Island, Año Nuevo Mainland, and Point Reyes, the three sites with long time series. Between colonies, the long-term mean arrival date (the hyper-mean across years, $\mu_a$) varied by 4.6 days: 10 Jan at the Channel Islands, 11 Jan at Año Nuevo Island, 14 Jan at Point Reyes, and 15 Jan at Año Nuevo Mainland and King Range (Fig 3A). The long-term mean date of the peak census was 15–17 days after mean arrival: 25 Jan at the Channel Islands, 26 Jan at Año Nuevo Island, 28 Jan at

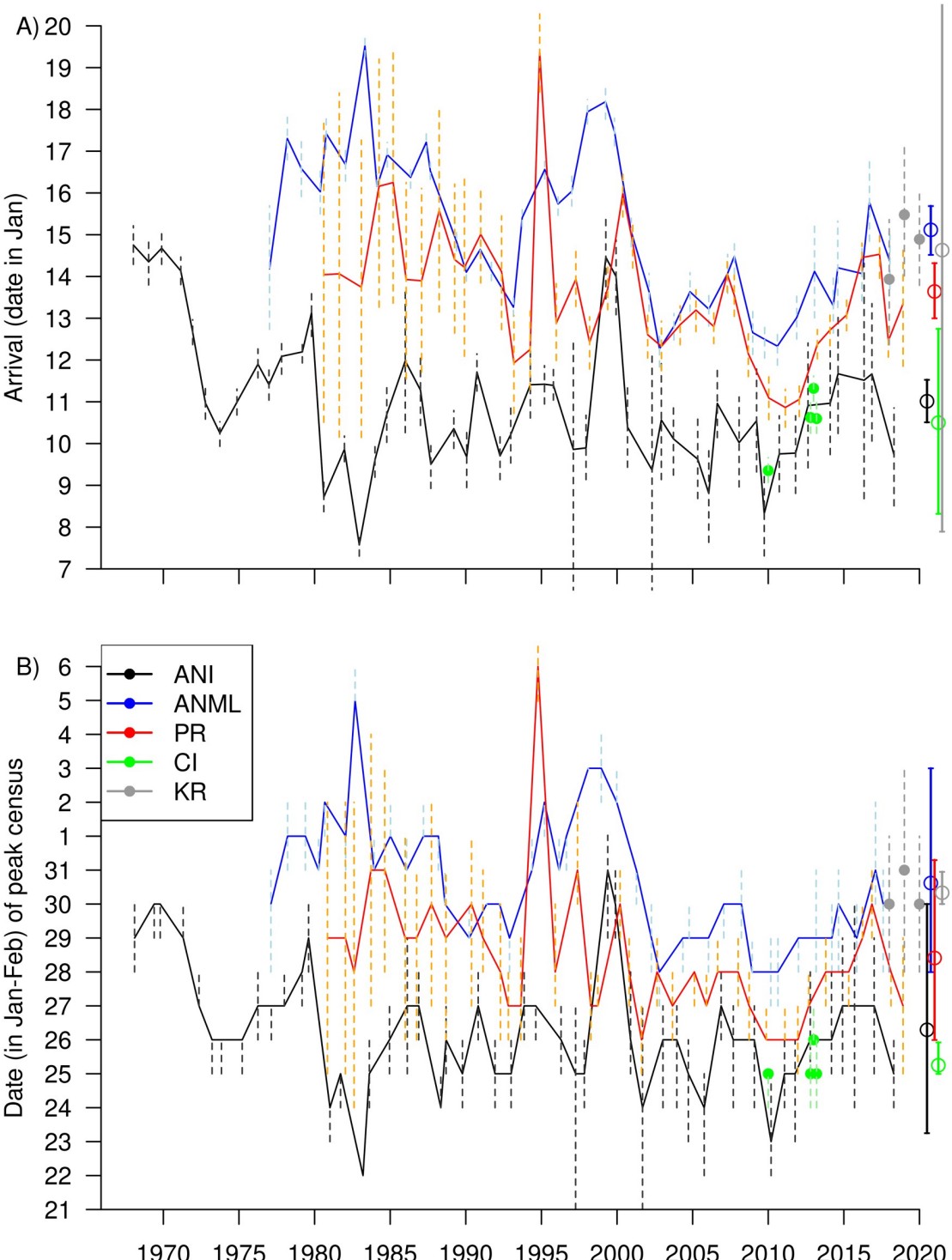

**Fig 3. Variation in census timing.** A) Mean annual arrival date, $\hat{a}_i$, and B) date of peak female census, both estimated from the census model. Three curves are from colonies with long time series (ANI: Año Nuevo Island; ANML: Año Nuevo Mainland; PR: Point Reyes); the four green points are individual years from the Channel Islands (CI), and the three gray points from King Range (KR). Dashed vertical lines show 95% credible intervals in each year. At the far right, open circles and solid lines show the hyper-means (long-term means) and credible intervals for each site.

Point Reyes, 30 Jan at King Range, and 31 Jan at Año Nuevo Mainland. Variation in the peak mirrored variation in arrival (Fig 3B).

Besides year-to-year fluctuations in arrival date or peak census date of < 2 days, there were a few large outliers (Fig 3). Some of the outliers were associated with unusual events. At Año Nuevo, 1983 had an enormous storm on 27 Jan, and surf covered the island beach. Most late-arriving females used the mainland, creating early arrival on the island and late on the mainland [19]. Again in 2010 at Año Nuevo and in 1995, 1998, and 2010 at Point Reyes, large storm surf forced females to move around, some to other colonies, thus creating unusual census patterns [20]. Other outliers were associated with poor census coverage, for example in 1998–2000 at Año Nuevo [4]. On the other hand, there was a steady delay in arrival over 2010–2015, consistent at four sites, supported by thorough census data and not associated with major storms. At the Channel Islands and King Range, the few years of data were consistent, but credible intervals were wide due to small samples (Fig 3).

### Correction factors for single counts

Using a single count at the peak of the census curve, the correction factor for yielding total population size was 1.13 at Año Nuevo Mainland, King Range, and the Channel Islands, 1.15 at Point Reyes, and 1.17 at Año Nuevo Island (A1-A5 Tables in S1 Appendix). A count at the peak had the narrowest credible intervals. Within ±2 days of the peak, correction factors and credible intervals barely changed, but beyond ±5 days, they increased rapidly (Fig 4). The correction curve was early at Año Nuevo Island and the Channel Islands compared to the other three locations, corresponding to differences in female arrival and thus census timing. At all five sites, the multiplier was between 1.13 and 1.20 from 26–30 Jan (Fig 4, Table 2).

## Discussion

We provide a method for estimating total population size in species whose presence in a study area is asynchronous. In these cases, the entire population is not present at the same time, so no direct count includes all animals [1]. This is common in pinnipeds and colonial birds. Here we demonstrated the method's effectiveness at describing asynchrony and estimating correction factors that account for animals not present during any one count. We found consistent correction factors in elephant seals, with the number of animals missed in the peak count close to 15% at all colonies. In other species, the fraction missed can be much higher [1, 3], but our approach should work regardless of the degree of synchrony. We provide software for testing it with any count data [21].

The complete population using a site is known as the superpopulation, and existing estimators are based on mark-recapture analysis [1, 3, 22]. These approaches are complemented by our method because the mark-recapture data can describe arrival and departure behavior and thus estimate the length of time individuals remain on site. Once site tenure is quantified, our model can be applied without any marks. The requirement for knowledge about tenure arises because two of the six parameters in our model must have prior estimates, otherwise the system is over-parameterized and abundance cannot be estimated. In elephant seals, we used marked animals to demonstrate that females remain on the colony for a mean of 31 days with a standard deviation of 4 days. Alternatively, the model could be adapted to cases where arrival or departure dates are known in advance but tenure is not. An extra detail in our model, a correlation between arrival and tenure, might be omitted in other species, since it may have little impact on population estimates.

Unlike the consistent timing in elephant seals, birth date in gray seals (*Halichoerus grypus*) shifted earlier by 18 days over 25 years [15]. Our model, however, can handle any variation in

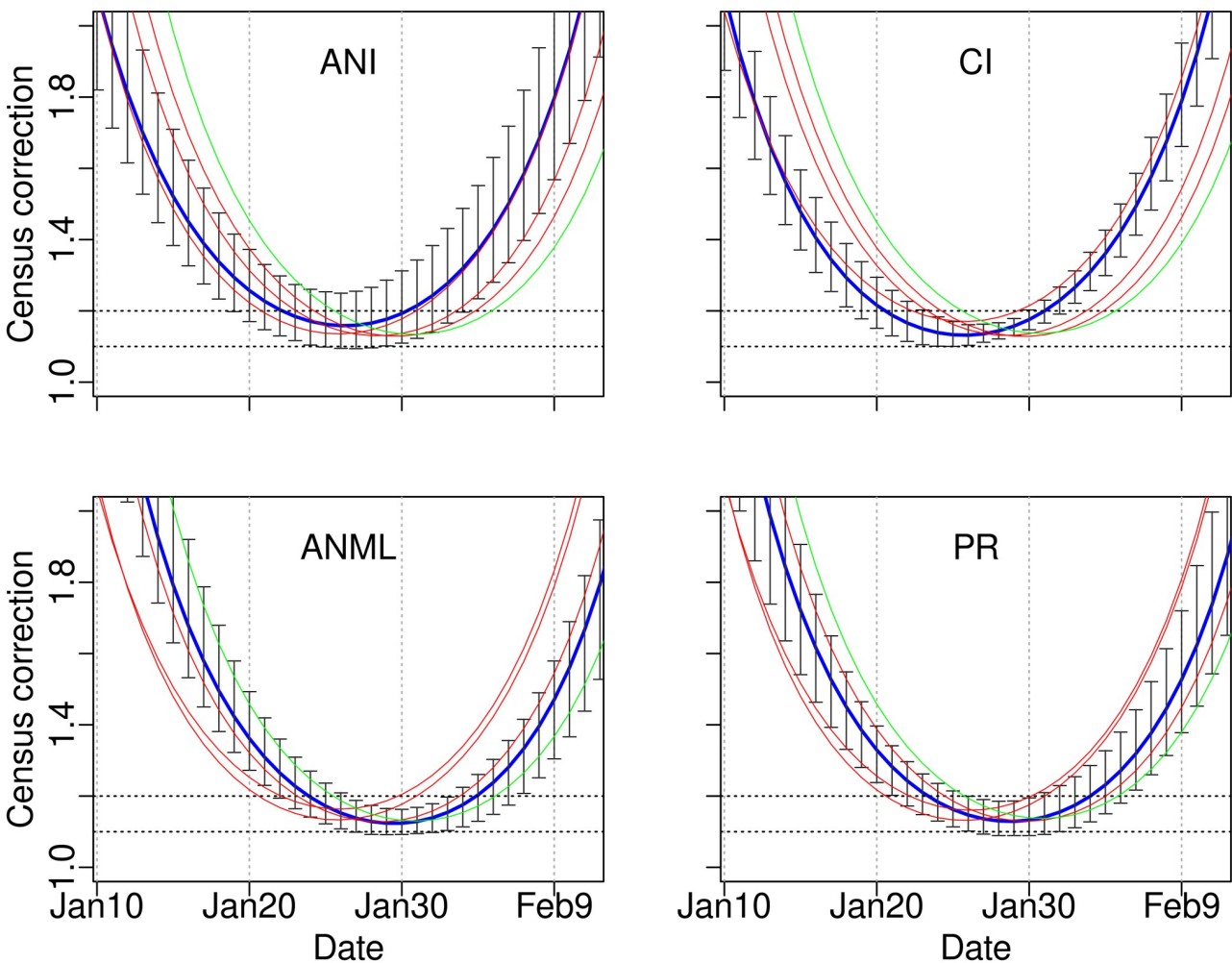

**Fig 4. Correction factors.** The model's estimated multiplier for converting any daily count into a total female population, as a function of date (ANI = Año Nuevo Island, ANML = Año Nuevo Mainland, PR = Point Reyes, CI = Channel Islands, KR = King Range). Each panel shows the curve for one of those four sites highlighted in blue, including credible intervals; the red lines are the remaining sites, with King Range added in green in all panels. Dotted horizontal guide lines are at 1.1 and 1.2. The census and hence the correction curve was earlier at Año Nuevo Island and the Channel Islands, later at the other three sites.

timing, indeed we demonstrated a shift toward earlier birth dates over the first few years at Año Nuevo and Point Reyes, though by only 4–5 days. Gray seals evidently responded to climatic variaton by pupping earlier [15], but we see no indication of climatic response in elephant seals. Instead, the slightly later breeding dates in recently-founded colonies are caused by animals assorting themselves. Timing is delayed in new colonies because late-arrivers at established colonies encounter high density and are likely to emigrate to new sites, especially during storm surge [19, 23–25]. It does not involve population-wide change in timing, such as must be happening in gray seals.

Had elephant seals shifted their timing as much as gray seals, correction factors would have to change through the years. But consistency through time and across colonies meant that 26–30 Jan were close to the optimal day for a count at all sites, and remained so for 30 years where long time series were available. Moreover, the correction factors at all sites were 1.13–1.20, with credible intervals 1.1–1.3. We thus recommend that counts at new colonies, where long-term data are lacking, should be 26–30 Jan, and would lead to a population estimate within

**Table 2. Correction factors for estimating the total breeding female population at five different sites (the Channel Islands site is three colonies combined).** The total population using the colony in a year is estimated by multiplying a count on any of the dates by the given factor. Credible intervals and a longer series of dates are provided the S1 Appendix. ANI = Año Nuevo Island, ANML = Año Nuevo Mainland, PR = Point Reyes, CI = Channel Islands, KR = King Range.

| Date | ANI | ANML | PR | CI | KR |
|---|---|---|---|---|---|
| Jan 23 | 1.19 | 1.23 | 1.21 | 1.15 | 1.29 |
| Jan 24 | 1.18 | 1.20 | 1.19 | 1.14 | 1.25 |
| Jan 25 | 1.17 | 1.17 | 1.17 | 1.13 | 1.22 |
| Jan 26 | 1.17 | 1.15 | 1.15 | 1.13 | 1.19 |
| Jan 27 | 1.17 | 1.14 | 1.14 | 1.13 | 1.17 |
| Jan 28 | 1.17 | 1.13 | 1.13 | 1.14 | 1.15 |
| Jan 29 | 1.19 | 1.13 | 1.13 | 1.16 | 1.14 |
| Jan 30 | 1.20 | 1.13 | 1.14 | 1.17 | 1.14 |
| Jan 31 | 1.23 | 1.13 | 1.15 | 1.20 | 1.14 |

10% error, excellent precision for estimating the population size of a highly pelagic marine predator.

In the closely related southern elephant seal, the census curve is similar [8], and the date of the female count is consistent across colonies at the core of the latitudinal range [9]. The far northern Peninsula Valdes colony, however, is 10 days earlier [26], and it is thus clear that a separate census correction for Valdes would be needed. We will need to address this issue with the northern elephant seal because the Mexican colonies breed earlier than the ones we studied in central California, and the optimal census in Mexico is probably 15–20 January [27].

The colonies we examined varied greatly in size. The Channel Islands each had over 10,000 females spread across many beaches, but at Año Nuevo, we separated the single beach on the island from the nearby mainland groups, and the King Range colony had only 100 animals. Despite these differences, the correction factors barely differed. We conclude that a population of breeding elephant seals can be estimated effectively from just one count, whether a single isolated group or a large number of groups counted together. Once the method is adapted to sites in Mexico, we believe that the entire range—six major colonies in the United States and two in Mexico [7, 28]—could be monitored with single counts per year, effectively estimating the world population of a deep-ocean predator with minimal investment of time and funds.

## Supporting information

**S1 Appendix. All supporting tables and figures, including full correction factors at five colonies, plus additional results for two subcolonies at the Año Nuevo mainland.**
(PDF)

## Acknowledgments

We thank Marshall Sylvan, who stated then solved the asynchrony problem in the 1970s at Año Nuevo Island. We also thank numerous scientists for field work and advice, especially J. Reiter, G. Oliver, R. Gisiner, M.O. Pierson, D. Adams, J. Longstreth, D. Notthelfer, J. Pettee, D. Press, S. Waber, S. Van Der Wal, B. Becker, M. Cox, J. Irwin, J. McAbery, P. Ruiz-Lopez, R. Hein, P. Forman, C. Nasr, and E. Levy. We thank the staff of Año Nuevo State Park for authorizing and coordinating research at Año Nuevo, and the Institute of Marine Sciences and the University of California, Santa Cruz, for field support at Año Nuevo Island.

## Author Contributions

**Conceptualization:** Richard Condit, Daniel P. Costa.

**Data curation:** Richard Condit, Sarah G. Allen, Sarah Codde, P. Dawn Goley.

**Formal analysis:** Richard Condit.

**Funding acquisition:** Sarah G. Allen, Daniel P. Costa, Sarah Codde, P. Dawn Goley, Burney J. Le Boeuf, Mark S. Lowry.

**Investigation:** Richard Condit, Sarah G. Allen, Daniel P. Costa, Sarah Codde, P. Dawn Goley, Burney J. Le Boeuf, Mark S. Lowry, Patricia Morris.

**Methodology:** Richard Condit, Sarah G. Allen, Sarah Codde, P. Dawn Goley, Burney J. Le Boeuf, Mark S. Lowry, Patricia Morris.

**Project administration:** Richard Condit, Sarah G. Allen, Daniel P. Costa, Sarah Codde, P. Dawn Goley, Burney J. Le Boeuf, Mark S. Lowry.

**Resources:** Sarah G. Allen, Daniel P. Costa, Sarah Codde, P. Dawn Goley, Burney J. Le Boeuf, Mark S. Lowry.

**Software:** Richard Condit.

**Supervision:** Richard Condit, Daniel P. Costa, Sarah Codde, P. Dawn Goley, Burney J. Le Boeuf.

**Validation:** Richard Condit, Sarah G. Allen, Sarah Codde, P. Dawn Goley, Mark S. Lowry, Patricia Morris.

**Visualization:** Richard Condit.

**Writing – original draft:** Richard Condit.

**Writing – review & editing:** Richard Condit, Sarah G. Allen, Daniel P. Costa, Sarah Codde, P. Dawn Goley, Burney J. Le Boeuf, Mark S. Lowry, Patricia Morris.

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
