## [Decision Letter · Decision Letter 0]

20 May 2021

PONE-D-21-03950

Estimating the size of a breeding colony of northern elephant seals from a single census

PLOS ONE

Dear Dr. Condit,

Thank you for submitting your manuscript to PLOS ONE. Firstly my apologies for the delay in the revision process, this was not through any fault of the manuscript. After careful consideration, we feel that it has merit but does not fully meet PLOS ONE’s publication criteria as it currently stands. Therefore, we invite you to submit a revised version of the manuscript that addresses the points raised during the review process.

Reviewer 1 has raised a number of concerns that should be considered in your response to ensure the manuscript meets PLOS ONE's publication criteria. However, while Reviewer 1 feels that the lack of generalisabilitiy of your method is a concern and a barrier for publication, this is not one of PLOS ONE's publication criteria. I will leave it to you as to how you wish to respond to this particular point, however, perhaps a discussion point that highlights the value of your model for your study species over a more general and, likely, less precise approach.

We look forward to receiving your revised manuscript.

Kind regards,

Andrew J. Hoskins

Academic Editor

PLOS ONE

Journal Requirements:

2.In your Data Availability statement, you have not specified where the minimal data set underlying the results described in your manuscript can be found. PLOS defines a study's minimal data set as the underlying data used to reach the conclusions drawn in the manuscript and any additional data required to replicate the reported study findings in their entirety. All PLOS journals require that the minimal data set be made fully available. For more information about our data policy, please see http://journals.plos.org/plosone/s/data-availability.

Reviewers' comments:

Reviewer's Responses to Questions

**Comments to the Author**

1. Is the manuscript technically sound, and do the data support the conclusions?

Reviewer #1: Partly

Reviewer #2: Yes

2. Has the statistical analysis been performed appropriately and rigorously? 

Reviewer #1: Yes

Reviewer #2: Yes

3. Have the authors made all data underlying the findings in their manuscript fully available?

Reviewer #1: Yes

Reviewer #2: Yes

4. Is the manuscript presented in an intelligible fashion and written in standard English?

Reviewer #1: No

Reviewer #2: Yes

5. Review Comments to the Author

Reviewer #1: The authors have developed a methodology to estimate the annual number of breeding female northern elephant seals at seven study colonies. An impressive dataset has been collected and analysed to accomplish this goal. The premise is simple and the results are of value to the field of population ecology. However, the delivery is poor. The narrative is not flowing, not enough background information is provided, the language alternates been informal and formal speech, and there many misuses of the English language. Most importantly, the authors have made the paper single-species focused rather than using their study species as a model organism to develop a methodology that can be applied to other polygynous, capital breeding species. This is not within the scope of PLoS ONE.

I found it extremely frustrating that no line numbers were provided in the manuscript. Given that the manuscript could only be downloaded in PDF format, I could not add my own referencing system.

Please could the authors address all of my comments and questions in the manuscript as well as in their written response to the reviewer.

Please refer to the attached document for detailed comments.

Reviewer #2: I enjoyed reading the paper which makes a really nice contribution to assessing population numbers. Accurate long-term counts of animals is central to quantifying population status and trends and for this growing population this is especially useful. The paper is statistically sound as are the methods, The results are well supported and their interpretation and discussion appropriate. I especially like that there are simple correction factors that can be used and applied.

6. PLOS authors have the option to publish the peer review history of their article (what does this mean?). If published, this will include your full peer review and any attached files.

Reviewer #1: **Yes: **Dr Kyle J. Lloyd

Reviewer #2: No

---

## [Author Response · Author response to Decision Letter 0]

16 Nov 2021

Following Reviewer 1's suggestion that the method be broadened to other species, we substantially rewrote the introduction and discussion to cover asynchronous populations in other species. Most important, we rewrote software to make it available for testing with other datasets. The original source code is published at github, and there is a web portal where data can be uploaded and the program executed, without need for the raw code.

Reviewer 1 also made many suggestions for improving the presentation, and we have followed most. We added background in the Introduction and Methods on basic elephant seal biology, and expanded the statement of the problem, the data available, and the goals of the model. The discussion was rewritten to include a broader context of population estimators in other species, and references were updated. We also removed Appendix 2, which is useful to us but, as noted by Reviewer 1, did not form part of the manuscript. Many other detailed suggestions were incorporated. Here are a few cases where we disagreed:

In some places, Reviewer 1 requested more details about the model, but it was published in full in 2007, and the earlier paper provides details on the parameters and how they are estimated. It would be redundant to repeat them in this paper.

Abbreviations were retained in the figures. In some space was tight, so they were used consistently throughout.

Reviewer 1 suggested we mention polygyny in elephant seals, and they certainly are polygynous. But the problem at hand is estimating the number of breeding females, and we have explained why males are less important (as the Reviewer suggested). But polygyny is not relevant to counts and modeling the population of breeding females.

We did not always agree with Reviewer 1's opinions about English usage. In some cases, the reviewer's preference is not the only correct usage, and we wrote what we think is best.

---

## [Decision Letter · Decision Letter 1]

20 Dec 2021

Estimating population size when individuals are asynchronous: a model illustrated with northern elephant seal breeding colonies

PONE-D-21-03950R1

Dear Dr. Condit,

We’re pleased to inform you that your manuscript has been judged scientifically suitable for publication and will be formally accepted for publication once it meets all outstanding technical requirements.

Kind regards,

Andrew J. Hoskins

Academic Editor

PLOS ONE

Additional Editor Comments (optional):

Reviewers' comments:

Reviewer's Responses to Questions

**Comments to the Author**

1. If the authors have adequately addressed your comments raised in a previous round of review and you feel that this manuscript is now acceptable for publication, you may indicate that here to bypass the “Comments to the Author” section, enter your conflict of interest statement in the “Confidential to Editor” section, and submit your "Accept" recommendation.

Reviewer #2: All comments have been addressed

2. Is the manuscript technically sound, and do the data support the conclusions?

Reviewer #2: Yes

3. Has the statistical analysis been performed appropriately and rigorously? 

Reviewer #2: Yes

4. Have the authors made all data underlying the findings in their manuscript fully available?

Reviewer #2: Yes

5. Is the manuscript presented in an intelligible fashion and written in standard English?

Reviewer #2: Yes

6. Review Comments to the Author

Reviewer #2: Thank-you for your thorough revsion, thisn is a useful paper and one that will be of interest to many of the PlosOne readers

7. PLOS authors have the option to publish the peer review history of their article (what does this mean?). If published, this will include your full peer review and any attached files.

Reviewer #2: No

---

## [Editor Report · Acceptance letter]

13 Jan 2022

PONE-D-21-03950R1 

Estimating population size when individuals are asynchronous: a model illustrated with northern elephant seal breeding colonies 

Dear Dr. Condit:

I'm pleased to inform you that your manuscript has been deemed suitable for publication in PLOS ONE. Congratulations! Your manuscript is now with our production department. 

Kind regards, 

on behalf of

Dr. Andrew J. Hoskins 

Academic Editor

PLOS ONE